# Psychological Burden during the COVID-19 Pandemic in Female Caregivers of Preterm versus Term Born Children

**DOI:** 10.3390/children10050787

**Published:** 2023-04-27

**Authors:** Maire Brasseler, Madeleine Fink, Barbara Mitschdörfer, Margarete Reimann, Eva-Maria Skoda, Alexander Bäuerle, Martin Teufel, Ursula Felderhoff-Müser, Britta Hüning

**Affiliations:** 1University Hospital Essen, Department of Paediatric I, Neonatology, Paediatric Intensive Care, Pediatric Neurology, University Duisburg-Essen, 45122 Essen, Germany; 2Centre for Translational Neuro- and Behavioral Sciences, C-TNBS, Faculty of Medicine, University Duisburg-Essen, 45122 Essen, Germany; madeleine.fink@lvr.de (M.F.);; 3Clinic for Psychosomatic Medicine and Psychotherapy, LVR-University Hospital Essen, University of Duisburg-Essen, 45147 Essen, Germany; 4Bundesverband “Das frühgeborene Kind e.V.”, 60598 Frankfurt am Main, Germany; 5Bunter Kreis, University Hospital Essen, University Duisburg-Essen, 45122 Essen, Germany

**Keywords:** preterm, COVID-19, anxiety, depression, pandemic, protection behavior, safety behavior

## Abstract

Background: during the COVID-19 pandemic, psychological burden increased. Contact restrictions were predominantly stressful for families. Parenthood was reported to be especially challenging for parents of preterm children. Material and Methods: a cross-sectional online-based survey on the psychological burden of parents of preterm and full-term born infants and toddlers during the second lockdown of the COVID-19 pandemic in Germany was offered by social media, webpages, etc. Generalized Anxiety Disorder-7 (GAD-7), Patient Health Questionnaire-2 (PHQ-2), COVID-19 Anxiety (C-19-A), COVID-19-related child protection behavior (PB) were used. Results: 2742 parents—predominantly females—took part in the study, 2025 parents of full-term and 717 parents of preterm born children. Female caregivers of full-term children reported significantly more depression symptoms than those of preterm children during the second lockdown of the COVID-19 pandemic. The PB correlated with increased COVID-19 anxiety as well as with increased generalized anxiety and depression symptoms. Female caregivers of preterm children showed significantly more protection behavior than those of full-term born children.

## 1. Introduction

Parental mental health contributes substantially to children’s cognitive, emotional, social and physical development [1,2]. Persisting parental depressive symptoms were shown to predict child’s dysfunction in follow-up studies of infants [3]. Multiple studies reported mental health problems in parents following preterm birth [4,5]. The often unexpectedly early delivery of their child and circumstances of the Neonatal Intensive Care Unit contribute to parental trauma [6] with a close correlation between birthweight and parental stress [5,7]. Mothers of preterm infants < 32 weeks gestational age are at high risk of posttraumatic stress 18 months following preterm delivery (prevalence of 60.4%) [8].

Stressful life events are events that are assessed as threatening by the individual and trigger a physiological reaction [9]. By this definition, COVID-19 could be considered a stressful life event that may lead to a psychological burden [9,10,11], defined as elevated symptoms of generalized anxiety, depression and overall COVID-19 anxiety. There is limited evidence on the impact of stressful life events, such as the pandemic and its social consequences on depressive and anxiety symptoms among caregivers of preterm infants. In addition, less is known whether multiple stressful life events such as preterm delivery and the circumstances of the COVID-19 pandemic increase the risk of disease [12].

In general, various systematic reviews and meta-analyses revealed the negative impact of the COVID-19-pandemic, isolation and quarantine on mental health [13,14]. In Germany, several studies showed a strong impact of the pandemic and related circumstances, prohibitions and restrictions [15,16,17,18,19]. Correlations between COVID-19 anxiety, depression and generalized anxiety symptoms were described [16].

The impact on mental health is even more pronounced in patients suffering from mental or chronic somatic illnesses [20,21]. These patients are at high risk for severe symptom exacerbation during the etiopathology in COVID-19 [18,22].

COVID-19 anxiety is defined as the specific phobia of COVID-19 and is measured by the COVID-19 anxiety scale [23] by a screening questionnaire measuring anxiety regarding COVID-19, a modification of the Specific Phobia Adult questionnaire (SP-D scale). COVID-19 anxiety influences the pandemic safety behavior [24,25], such as social distancing and hand washing, and dysfunctional behavior, e.g., hoarding and stockpiling [25,26,27,28]. Differences in safety behavior depend on psychosocial and sociocultural determinants in the context of the pandemic [29]. Earlier studies analyzed the self-protection behavior and the influence of psychological burden on it. It is reasonable to assume that parental psychological burden also affects their child’s protection behavior, such as hygiene strategies to limit the risk of infection.

Parents faced particular challenges during the pandemic, partly due to the responsibility for their child and family’s health. Social distancing, closure of schools, day care and leisure facilities significantly increased parental stress [30]. The authors reported a very high burden of depression and anxiety in subgroups of parents.

The present study was conducted to evaluate the psychological burden in female caregivers in Germany during the second lockdown in 2020/21. Affective symptoms and associations of generalized anxiety and depression symptoms, COVID-19 anxiety and child protection behavior of female caregivers of term and preterm born children were determined. The hypothesis of the study was that caregivers of preterm children report a higher psychological burden and higher child COVID-19-related protection behavior.

## 2. Materials and Methods

### 2.1. Study Design

This cross-sectional study was performed during the second lockdown of the COVID-19 pandemic from 18 December 2020 until 15 March 2021 in Germany. A population-based online survey was conducted and distributed via social media (homepages, Facebook, Instagram) and the parent’s representative’s organization Bundesverband “Das frühgeborene Kind e.V.”. The study was approved by the Ethics Committee of the Medical Faculty of the University of Duisburg-Essen (20-9307-BO) and conducted in accordance to the Declaration of Helsinki.

The study addressed parents who had custody of a child younger than 18 years. The following socio-demographic data was evaluated: age of mother and child, education, marital status, occupation, residential details, childcare and health status. The survey distinguished between parents of preterm (gestational age < 37 weeks) and term born children (gestational age ≥ 37 weeks). Three validated measures were used to assess parental burden (Generalized anxiety disorder 7 (GAD-7) [31], Patient Health Questionnaire-2 (PHQ-2) [32], COVID-19-Anxiety Questionnaire (C-19-A) [23]). In addition, questions for the assessment of child protection behavior (see Appendix A) and preterm birth-associated diseases were drawn up with experts and parents’ representatives to reach consensus.

The completion of the study required about 15–20 min. Participation was voluntary and anonymous. Participants could exit the survey at any time.

### 2.2. Assessments

Generalized anxiety symptoms such as extreme fear, excessive worry about everyday things and being highly nervous about everyday circumstances [33] were measured by the GAD-7. The GAD-7 assessed the frequency of symptoms of anxiety during the last two-week period and consisted of seven items. A 4-point Likert scale was used for each item. A total score of ≥5, ≥10 and ≥15 indicates mild, moderate, or severe generalized anxiety symptoms, respectively [31]. The internal consistency was high with a Cronbach’s α of 0.901 [31].

Depression symptoms were defined by the diagnostic and statistical manual of mental disorders [34]. Depression symptoms were determined by the PHQ-2 which consists of two items. The frequency of depression symptoms during the last two-week period was measured by a Likert scale. A sum score ≥3 indicates major symptoms of depression [32]. Cronbach’s α was 0.767, which indicated a high internal consistency [32].

The COVID-19-Anxiety Questionnaire is a modification of the DSM 5–Severity Measure for Specific Phobia-Adult scale. The original descriptions were changed to COVID-19 specific phrases [23]. The COVID-19-Anxiety Questionnaire consists of all ten items of the Specific Phobia-Adult scale. There was no modification of item 7 and 10. The C-19-A scale showed an internal consistency of α = 0.86 [23].

The child COVID-19-related protection behavior questionnaire consisted of 11 items (Appendix A). The scale was conducted by expert consensus and in cooperation with the parents’ representatives. A 7-point Likert scale was used for each item (ranging from “1 = strongly disagree” to “7 = strongly agree”). The internal reliability of this study was high with Cronbach’s alpha 0.733. We conducted a mean for the scale.

### 2.3. Statistical Analysis

Data analysis was performed by SPSS Statistics 27 Software (IBM, Armonk, NY, USA). Reliability testing was conducted for the COVID-19-related child protection behavior items and a mean for the scale was computed. Mean sum-scores of the GAD-7, PHQ-2 and C-19-A were calculated for the whole group.

In subgroup analysis the GAD-7, PHQ-2 and C-19-A were determined separately for the group of female caregivers with preterm children and the group of participants with term born children. In addition, a descriptive subgroup analysis was performed for parental age, marital status, family role, education level, previously diagnosed psychiatric disorder, city size population, occupation, child’s age, child’s disease and gestational age. Due to the absence of a normal distribution, a Mann-Whitney U Test (Monte Carlo) was conducted in case of ordinal distribution. The level of significance in the test was set at 0.05. Furthermore, calculations of correlation were performed of child protection behavior, GHD-7, PHQ-2 and C-19-A and a multiple regression was tested for child protection behavior and GHD-7, PHQ-2, C-19-A, parental age, children age, education level of the parent, employer status and number of siblings. A priori sample size analysis was calculated for the multiple regression analysis by G*Power 3.1.9.7 (written by Franz Faul, University, Kiel, Germany) for a mediate effect size (f^2^ = 0.15) and a power (1-β error) = 0.95. For the eight predictors, a sample size of 160 and a critical F of 2.0 were computed.

## 3. Results

The survey was accessed 2742 times during the study period, 2701 female and 41 male caregivers/custodians participated in the survey. The survey was completed by 701 female caregivers of preterm children and 1991 female caregivers of full-term children (2693). A summary of inclusion criteria is presented in Figure 1. Data showed that 64.5% of the participants were 25–34 years old, 88.9% were married or lived in a relationship, 63.5% had a higher or university qualification and 64% had a child aged 0–3 years. For further details see Table 1. Only the records of female participants were used for further data analysis to account for gender differences. Because of the exclusion criteria, only results of parents 18 years and older were included.
Prevalence of generalized anxiety and depression symptoms and COVID-19 anxiety in female caregivers during the second lockdown in Germany

The prevalence of generalized anxiety and depression symptoms, COVID-19 anxiety and child protection behavior are shown in Table 2. Female caregivers of full-term children reported a significantly higher level of depression symptoms than those of preterm children during the second lockdown (*p* = 0.023, *Z* = −2.275, *N* = 2376).

Female caregivers of preterm children reported equal generalized anxiety symptoms as female caregivers of full-term children, where the mean prevalence was 6.44 (*SD* 5.04), respectively 6.63 (*SD* 4.89).

The mean prevalence of COVID-19 anxiety in female caregivers during the second lockdown was 18.84 (*SD* 5.90). Compared to the results of the normalized cohort for original SP-D scale in patients, the mean prevalence of COVID-19 anxiety was elevated in our study. To determine the COVID-19 anxiety, the results of 6262 participants were evaluated by Petzold et al. in 2020 and a mean score of 10.14 was reached in the general population [23].

The results of child protection behavior significantly differed in female caregivers of preterm and full-term children (Mann-Whitney-U-test). Female caregivers of preterm children reported significant higher child protection behavior than those of full-term children (*p* = 0.026, *Z* = −2.233, *N* = 2376).
Correlation of child protection behavior with generalized anxiety symptoms, depression symptoms and COVID-19 anxiety

The mean COVID-19-related child protection behavior was 30.78 (*SD* 9.67). It was positively correlated with generalized anxiety symptoms, depression symptoms and COVID-19 anxiety. The Spearman’s Rho and two-sided significance are shown in Table 3.
Multiple regression analysis

Child protection behavior is predicted by generalized anxiety and COVID-19 anxiety symptoms with a level of significance of ≤0.001. In addition, the female caregivers’ age and education level determine the child protection behavior with a lower level of significance of 0.021 and 0.032. The results of multiple regression analysis are shown in Table 4.
Subgroup analysis

In addition, the average prevalence of generalized anxiety, depression, COVID-19 anxiety and child protection behavior were calculated for the varying groups of the study population. The results of the descriptive analysis are shown in Appendix A. While female caregivers aged 18 to 24 years reported greater depression symptoms, those between 45 and 54 years revealed more child protection behavior and higher levels of generalized anxiety and COVID-19 anxiety.

Furthermore, widowed female caregivers reported more symptoms of generalized anxiety and COVID-19 anxiety. Participants with lower secondary education level had greater depression, generalized anxiety and COVID-19 anxiety symptoms than those with a higher education level. In case of non-employed status, higher depression symptoms were reported (Mann-Whitney U test: PHQ-2: *p* ≤ 0.001, *Z* = −3.425). The prevalence of generalized-anxiety symptoms, depression symptoms and COVID-19 anxiety was higher in mothers than in foster mothers (mothers: GAD-7: mean 6.59, *SD* 4.93; C-19-A: mean 18.85, *SD* 5.90, PHQ-2: mean 1.84, *SD* 1.66; foster mothers: GAD-7: mean 2.50, *SD* 3.54; C-19-A: mean 18.00, *SD* 5.66, PHQ-2: mean 1.00, *SD* 1.41) while foster mothers reported increased protection behavior (mothers: PB: mean 30.78, *SD* 9.67; foster mothers: PB: mean 29.50, *SD* 12.02) with non-significant differences. Increased psychological burden was noted in female caregivers with a previously diagnosed psychiatric disorder (Mann-Whitney U test: GAD-7: *p* ≤ 0.001, *Z* = −9.851; PHQ-2: *p* ≤ 0.001, *Z* = −8.768; C-19-A: *p* ≤ 0.001, *Z* =−4.684).

Female caregivers of preterm children with a gestational age < 32 weeks reported fewer symptoms of depression, generalized anxiety and COVID-19 anxiety than participants of preterm children with a gestational age of 32 to 36 weeks. When the child suffered from bronchopulmonary dysplasia, asthma or had surgery, their caregiver’ child protection behavior was significantly higher (Mann-Whitney U test: PB: bronchopulmonary dysplasia: *p* = 0.001, *Z* = −3.186; asthma: *p* = 0.008, *Z* = −2.647; after surgery: *p* = 0.013, *Z* = −0.486). The prevalence of generalized anxiety (*p* = 0.015), depression (*p* = 0.004) and COVID-19 anxiety symptoms (*p* < 0.001) were noticeably increased when the child suffered from asthma (Mann-Whitney U test: asthma: GAD-7: *p* = 0.015, *Z* = −2.423; PHQ-2: *p* = 0.004, *Z* = −2.842; C-19-A: *p* ≤ 0.001, *Z* = −3.408). All four measured parameters were lower in female caregivers of children born during the pandemic (aged 12 months or less) than the reported prevalence in female caregivers of children born before. Differences of generalized anxiety symptoms and depression symptoms were significant (Mann-Whitney U test: GAD-7: *p* ≤ 0.001, *Z* = −5.722; PHQ-2: *p* ≤ 0.001, *Z* = −4.353; C-19-A: *p* = 0.006, *Z* = −2.750; PB: *p* = 0.012, *Z* = −2.526).

## 4. Discussion

In the present study, generalized anxiety, depression and COVID-19 anxiety as well as child protection behavior were measured in female caregivers of preterm and full-term born children in Germany during the second lockdown of the pandemic, a potential stressful life event. This is the first study exploring changed behavior, child-related pandemic safety behaviors and protective behaviors in relation to mood and affective symptoms, respectively.

Our results disprove the hypothesis that female caregivers of preterm children more frequently report increased COVID-19 anxiety, symptoms of depression and generalized anxiety than mothers of full-term born children. These results are surprising at first sight. In general, parents of preterm children are at greater risk of mental health problems and posttraumatic stress caused by sudden birth, circumstances on Neonatal Intensive Care Unit and children’s physical factors such as development and low birth weight [4,5,6,7,8]. However, the results should be considered in light of the support offered to families. The majority of participants had children younger than 3 years of age, a period with regular contacts to pediatricians and other professionals under normal circumstances. Around birth and after discharge, parents of very immature infants often receive psycho-social support and assistance in building a network of care providers as recommended by pediatric societies, parent organizations and the German federal committee (G-BA) [35,36,37]. Perceived social support has a positive effect on well-being in stressful life situations [38]. In addition, mothers of preterm children have acquired coping strategies that might be useful in these situations [39]. “Resilience mediated the effects of stressful COVID-19-related events on” mental health such as “depressive and anxiety symptoms” [40]. Overall, functional coping strategies were associated with increased well-being during the COVID-19 pandemic [41]. Further published studies reported that greater self-efficacy and perceived control are associated with psychological resilience [42,43]. Greater self-efficacy and control through postpartum education may account for a lower psychological burden.

During the COVID-19 pandemic, social distancing and lockdowns were reported to have had an impact on the psychological health of expectant mothers and new parents [44]. Parents expecting a child during the COVID-19 pandemic also faced constrained access to resources and strict rules with limitations in antenatal care and during birth [45]. The study by Aydin et al. describes that mothers giving birth during the pandemic in England reported more uncertainties related to birth, poor communication and increased feelings of anxiety and high levels of negative emotions [45]. In the two studies, no distinction was made according to the gestational age of the children. In contrast to the UK study, our results revealed a significantly lower prevalence of generalized anxiety and depression symptoms if the child was born during the pandemic. While the UK study was conducted from July 2020 to March 2021, we analyzed results of the second wave in Germany (12/20–03/21). The second wave resulted in a “lockdown light”, vaccines had already been used and more information on the virus existed [46] resulting in a higher grade of control and less psychological burden [16].

Our results show that female caregivers of preterm children show greater child protection behavior than those of full-term born children. As recommended by professionals and parent organizations (such as European foundation for the care of newborn infants), parents of preterm born infants readily learn rules for hygiene and safety behavior to reduce the risk of infections due to a weakened immune system (e.g., sepsis) on Neonatal Intensive Care Units [47]. Parents of preterm children are aware of the higher risk for respiratory diseases [48,49]. Female caregivers of children with asthma or a bronchopulmonary dysplasia reported significantly higher child protection behavior. Since most of the participants had a higher education level, the benefit of increased safety behavior dealing with a virus such as SARS-CoV2 attacking the respiratory system is obvious.

Child protection behavior positively correlates with the prevalence of generalized anxiety symptoms, depression symptoms and COVID-19 anxiety. This is the first report of these correlations. Earlier studies reported correlations of generalized anxiety, depression symptoms, self-protection behavior and safety behavior, respectively [16]. Even if female caregivers of preterm children reported the same symptoms of generalized anxiety and COVID-19 anxiety as those of full-term children, their child protection behavior was higher in our study, most likely due to the internalized hygiene rules.

Not only can maternal mental health lead to altered perceptions of the child’s social behavior, but it can also lead to parental overprotection [50,51]. Greene et al. were able to predict parental perceptions of child vulnerability through maternal anxiety after preterm birth [51]. This “vulnerable child syndrome” includes abnormal separation difficulties, overindulgence, sleep problems and long-term developmental problems such as lower language scores with 20 month corrected age, poor peer relationships and behavioral issues. Thus, clinicians should be attentive to the additional stresses in parents and parental perception of their child’s vulnerability and behavior.

Female caregivers of teenagers reported higher generalized anxiety symptoms, COVID-19 anxiety and depression symptoms than female caregivers of children aged 0–10 years. Explanations may be found in the changing attachment behavior of adolescents —away from the parents, towards the peer group. Thus, parental influence is limited. Adolescence is characterized by marked developmental changes in cognitive and social skills. Adolescents show more risk-taking behavior due to different developmental rates of the frontal brain and the amygdala [52].

The fact that female caregivers of preterm children seem to be less affected could be due to the fact that preterm born children and adolescents more often show personality traits such as anxiety and shyness. Thus, they are often more insecure and reserved in social contact [53]. Peer interactions and social networks correlate with the gestational age. Children with lower gestational age often have a smaller social network [54]. In the context of COVID-19, lower social contacts may be reassuring for parents.

Depression symptoms seem to be influenced by the gestational age of the child. Female caregivers of preterm children < 32 gestational age weeks reported fewer symptoms of depression than those of children with a higher gestational age. In 2010, Silverstein et al. reported an association of maternal depression with negative perceptions of children’s social abilities and decreased participation in preschool activities following very preterm birth [50]. In contrast, before the pandemic, this association was not significant among term born children. It remains speculative whether this effect is also evident in increased depressive symptomatology in female caregivers of term-born children due to the pandemic. However, consequences for the social-emotional development may be one reason why support of families with term and near term born children could be considered.

It is possible that female caregivers of preterm children have a higher level of anxiety related to the prematurity of their child than due to the COVID-19-pandemic. In addition, a positive effect of the German psycho-socio-support system should be considered as possible reason for a greater resilience in caregivers of preterm children. It is important to note that parental distress is a well-established risk factor for mental health problems in children [55,56,57] and child development in general.

Depressive symptoms of female caregivers were slightly increased compared to reports from females before the pandemic [58]. The average depressive symptoms score for women was 1.00 in Germany in 2006 [58].

The prevalence of generalized anxiety and COVID-19 anxiety symptoms did not differ significantly between groups. However, in the cohort of female caregivers, the mean value of COVID-19 anxiety was 18.84 vs. 10.14 reported by Petzold et al. in the general population at the beginning of the pandemic [23]. These average values for COVID-19 anxiety (mean 10.14) in the general population were already higher than those reported for original SP-D scale in patients seeking out-patient treatment for a mental disorder (mean 8.2) in Germany in 2010 and 2011 [23].

Increased generalized anxiety symptoms of female caregivers in Germany during the second lockdown period compared to the prevalence pre-pandemic were reported with 14.0% and 15.4% in preterm and term born children, respectively, compared to 6% in the general population pre-pandemic [31,59].

There are several limitations of this study: there was no baseline data pre-pandemic and the study is a cross-sectional and not longitudinal. Only 43 male caregivers participated in the survey, therefore only the records of female participants were used for further data analysis to account for gender differences. In general, females more frequently reported symptoms of depression and generalized anxiety during the COVID-19-pandemic [15,60]. Most of the participants (83.7%) of this study live in West Germany. In addition, an online survey was used to collect the data, which was distributed via online and analogue channels. Thus, a selection bias occurred, since the survey was promoted via social media channels, friends and peers, often from the same social environment.

However, this is the first study investigating female caregivers of preterm and full-term born children with 64.1% of participants having children < 4 years of age. Our results suggest a possible role for the supporting system of caregivers with preterm born children, especially in the first years of the child. Pediatricians should be aware of possible life-altering circumstances, e.g., stressful life events, and recognize the need for support in coping strategies in caregivers of full-term children. Online tools like “CoPE it” could be beneficial in such situations [61].

## Figures and Tables

**Figure 1 children-10-00787-f001:**
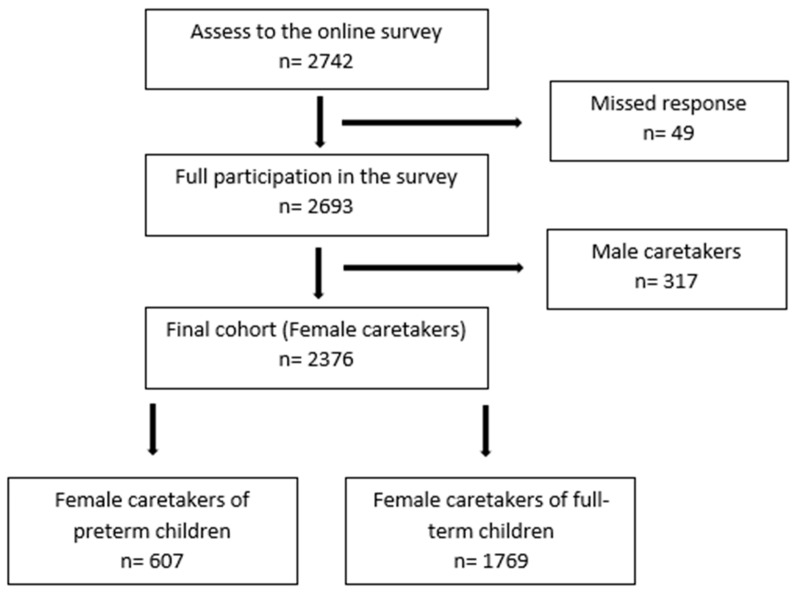
Flow chart of participants’ selection.

**Table 1 children-10-00787-t001:** Overview of the demographic participant’s characteristics.

	*N*	%
Age		
18–24 years	168	6.2
25–34 years	1736	64.5
35–44 years	723	26.8
45–54 years	63	2.3
55–74 years	3	0.1
Marital status		
Single	219	8.1
Married	1872	69.5
In a relationship	522	19.4
Divorced/separated	73	2.7
Widowed	3	0.1
Other	4	0.1
Family role		
Mother	2657	98.7
Foster mother	7	0.3
Others	25	0.9
Educational level		
University education	856	31.8
Higher education entrance		
Qualification	854	31.7
Secondary education	827	30.7
Lower secondary education	136	5.1
No qualification	5	0.2
Other	15	0.6
Occupation		
Not employed	609	22.6
Child’s age (Medium age: 2, 6 years)		
0–3 years	1726	64.1
4–6 years	331	12.3
7–10 years	190	7.1
11–17 years	86	3.2
Missing birth date	360	13.4
Childcare		
Family	1283	54.0
Others	1	0.0
Preschool	827	34.8
School	266	11.2
Breast feeding last 8 month		
Yes	898	37.8
Gestational age of preterm children		
32–36 weeks	345	56.8 *
28–32 weeks	173	28.5 *
≤28 weeks	89	14.7 *
Child’s diseases		
Bronchopulmonary Dysplasia	55	9.1 *
Operation	131	5.5
Asthma	74	3.1
Total	2693	100.0

* Of preterm children.

**Table 2 children-10-00787-t002:** Prevalence of symptoms of generalized anxiety symptoms (GAD-7), depression symptoms (PHQ-2), COVID-19 anxiety (C-19-A) and child protection behavior.

	*N*	Generalized Anxiety (GAD-7)	Depression (PHQ-2)	COVID-19 Anxiety (C-19-A)	Child Protection Behavior (PB)
Female caregivers of preterm children	607	6.44 [0.0–21.0]	1.70 [0.0–6.0]	18.83 [9.0–39.0]	31.65 [9.0–63.0]
		42.5% ^b^ 14% ^c^ 9.7% ^d^	22.1% ^e^		
Female caregivers of full-term children	1769	6.63 [0.0–21.0]	1.89 [0.0–6.0]	18.84 [1.0–43.0]	30.48 [9.0–61.0]
		35.1% ^b^ 15.4% ^c^ 8.7% ^d^	27.8% ^e^		
*p* (95% CI) ^a^		0.178	≤0.05	0.945	≤0.05
*Z*		−1.346	−2.275	−0.069	−2.233

Values shown as mean [minimum–maximum] or percentage (%). ^a^ Mann-Whitney U Test Monte Carlo.^b, c, d^ Cut off values of generalized anxiety, ^b^ ≥5 = mild, ^c^ ≥10 = moderate, ^d^ ≥15 = severe elevated symptoms of generalized anxiety. ^e^ Cut off values of depression, ≥3 = elevated symptoms of depression.

**Table 3 children-10-00787-t003:** Correlation of child protection behavior with generalized anxiety symptoms (GAD-7), depression symptoms (PHQ-2) and COVID-19 anxiety (C-19-A).

Correlation of Child Protection Behavior with	Spearman-Rho	*p*	
Generalized anxiety (GAD-7)	0.327	≤0.001	
Depression (PHQ-2)	0.237	≤0.001	
COVID-19 Anxiety (C-19-A)	0.422	≤0.001	

*N* = 2376.

**Table 4 children-10-00787-t004:** Multiple regression analysis to predict the child protection behavior by generalized anxiety and COVID-19 anxiety.

Predictor	*Β*	*Βse*	*T*	*p*
Generalized anxiety (GAD-7)	0.225	0.113	3.586	≤0.001
Depression (PHQ-2)	−0.085	−0.014	−0.501	0.617
COVID-19 anxiety	0.635	0.380	14.775	≤0.001
Female caregiver’ age	0.804	0.049	2.303	0.021
Child age	−0.000017	0.000	−0.18	0.985
Education level of female caregiver	0.482	0.046	2.147	0.032
Employer status	0.142	0.011	0.515	0.606
Number of siblings	0.371	0.029	1.352	0.177

Total R^2^ = 0.211 (*p* ≤ 0.001; *N* = 2376).

## Data Availability

The data presented in this study are available on request from the corresponding author.

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
