# Peer review of "Psychological Burden during the COVID-19 Pandemic in Female Caregivers of Preterm versus Term Born Children"

_children, 2023, doi:10.3390/children10050787_

Round 1
Reviewer 1 Report
Thank you for the opportunity to review your study. I want to start by acknowledging that your study is unique in looking at the impact of the COVID-19 pandemic on female caregivers with children who were born preterm versus term.
With that being said, I was not convinced you fully achieved this goal in the way you presented your study. First, you write in the abstract and conclusions that this study looked at "mothers." However, in the results, you keep saying "parents." Then when I look at your study data, it appears that "mother" doesn't necessarily fit all your study participants. It would be more accurate to say "female caregivers" and then maybe specify the percentage that indicated they were either a mother/foster mother. I would imagine there could be differences between birth mothers with other female caregivers, particularly if you are examining preterm versus full term.
Then I had made the assumption when I read your title, abstract, and even intro that these were going to be mothers who gave birth during the pandemic of pre-term versus term babies. This does not seem to actually be what you studied, though. Based on my reading of the study participants, many of them birthed their babies prior to the pandemic. If that is the case, I want to see you all re-word your conclusion to acknowledge the fact that these young children were birthed prior to the pandemic. I don't think you can conclude mothers of term babies had higher levels of reported depression because of shorter hospital stays if these happened pre-pandemic. I don't know if we can really ascertain any specifics as why this might be based on what you collected.
I am also curious as to why you chose the PHQ-2 and not the PHQ-9. I would imagine you could have gotten possibly better variability with the PHQ-9.
I am in agreement about the limitations of your study. I think if you rewrite your conclusions as well as results you would have a stronger study for publication.
Author Response
We thank the reviewer for the time he investigated and the important comments to improve our manuscript. We attached a point by point response as a word document.

Reviewer 2 Report
1. The background presented in the first four paragraphs of the introduction is too macro and lengthy to see how it relates to this paper. Too much information on the history and impact of the COVID-19 epidemic, especially now that it is being managed on a regular basis, will not engage the reader. It is recommended that the focus be on the groups of concern to this study.
2. How can you tell that most of mothers’ emotional changes stem from her children and not from other psychosocial factors such as work? Did this study control for other variables during the comparative analysis?
3. The practice recommendations given in this study did not distinguish between parents of preterm and full-term born children. It is recommended that the practice contribution be written in relation to the aims and conclusions of this study.
4. Please add the most recent literature since 2022 and describe the relevance of this study in today's social context. In other words, what can a focus on the psychological burden of mothers during COVID-19 offer to health governance nowadays?
5. Please use the statistical calculation tool to show that the sample size for this study is adequate.
6. What were the expectations of this study as mentioned in paragraph 11 of the Discussion section. The expectations need to be clearly stated in the preceding text and then compared with the results of the study.
7. The format of the figures in the tables is inconsistent and incorrectly labelled. For example, "1.726" in Form 2
8. The timing of the research for this study is time-sensitive. Please analyze this in the discussion section in relation to the timing of the research (i.e. 2020.12.18 - 2021.03.15).
Author Response
We thank the reviewer for the important review. The manuscript benefits substantially.

Reviewer 3 Report
Thank you for your effort in conducting this valuable and original study. There are few notes that should be addressed before going to the next step in publishing this manuscript.
There are 3 major issues that should be treated carefully.
1. The term psychological burden (in the title): the authors did not address what they meant by this term, also, the introduction does not introduce this term. This term is first appeared in line 204. Therefore, this term should be described carefully and included the major study variables (depression, anxiety, and protection behaviour).
2. The participants: there is a big confuse to the reader between the frequently used terms in the manuscript (mothers or parents). In all parts of the manuscript those are confusing terms; sometimes used mothers and sometimes used parents or custodians.
3. The sample size. in some places it is 2742, in others, 2692, and in another places is 2367. This discripancy confuse the reader. You have to be clear about the total sample size, mothers, fathers, parents of preterm, parents of full-term, mothers of preterm, and so on.....
There are other minor notes
1. Some statements and sentences are confusing and needing re-write: proof read may solve the language problem. (ex: paragraph lines 53-55).
2. not recommended to start a sentence with number (ex: line 148, 701), also not recommended to have two numbers with any separation (ex: line 77, 60.4% 18).
3. Assessment (page 3): Need to describe more about the COVID-19 anxiety; scoring system, composition of how many items as you did in other tools.
4. Reported Chronbach's alphas in lines 113, 117, 121, and 126; it is not clear if this alpha for this study or other studied (if other studies needs citations).
5. Report results of GAD-7 before line 156. you have reported in the text the results of depression, covid-19 anxiety, and protection behaviour and missed the first concept (GAD-7).
6. In table 3, why N for parents of preterm children is 607 not 701, and the same thing for full term (why 1769 not 1991).
7. Also, because it is confusing, no need in table 3 to have the cut-off points for GAD-7 and PHQ-2 (it is clear in the methods and it appears also at the end of the table).
8. be careful about using coma and period in the numbers. (ex: table 5, p is 0.021 or 0,021 and so on...).
9. in multiple regression line 184, are they two predicting variables or 4? What about parental age (0.021) and educational level of parents (0.032)?
10. conclusions and recommendations are required to strengthen the study.
Author Response
We thank the reviewer for the important comments and the time he investigated to improve the manuscript. We attached a point by point response as a word document.

Round 2
Reviewer 1 Report
Thank you for the opportunity to review your revised article. It is much improved from the original article you submitted. I appreciate the time you took to address each of the concerns expressed by the reviewers. I am much more comfortable now with recommending your article for publication.
Reviewer 2 Report
Accept